# Prevalence and characteristics of long COVID in elderly patients: An observational cohort study of over 2 million adults in the US

Kin Wah Fung◉*, Fitsum Baye, Seo H. Baik◉, Zhaonian Zheng, Clement J. McDonald◉

Lister Hill National Center for Biomedical Communications, National Library of Medicine, National Institutes of Health, United States of America

* kfung@mail.nih.gov

## Abstract

### Background

Incidence of long COVID in the elderly is difficult to estimate and can be underreported. While long COVID is sometimes considered a novel disease, many viral or bacterial infections have been known to cause prolonged illnesses. We postulate that some influenza patients might develop residual symptoms that would satisfy the diagnostic criteria for long COVID, a condition we call "long Flu." In this study, we estimate the incidence of long COVID and long Flu among Medicare patients using the World Health Organization (WHO) consensus definition. We compare the incidence, symptomatology, and healthcare utilization between long COVID and long Flu patients.

### Methods and findings

This is a cohort study of Medicare (the US federal health insurance program) beneficiaries over 65. ICD-10-CM codes were used to capture COVID-19, influenza, and residual symptoms. Long COVID was identified by (a) the designated long COVID code B94.8 (code-based definition), or (b) any of 11 symptoms identified in the WHO definition (symptom-based definition), from 1 to 3 months post-infection. A symptom would be excluded if it occurred in the year prior to infection. Long Flu was identified in influenza patients from the combined 2018 and 2019 Flu seasons by the same symptom-based definition for long COVID. Long COVID and long Flu were compared in 4 outcome measures: (a) hospitalization (any cause); (b) hospitalization (for long COVID symptom); (c) emergency department (ED) visit (for long COVID symptom); and (d) number of outpatient encounters (for long COVID symptom), adjusted for age, sex, race, region, Medicare-Medicaid dual eligibility status, prior-year hospitalization, and chronic comorbidities. Among 2,071,532 COVID-19 patients diagnosed between April 2020 and June 2021, symptom-based definition identified long COVID in 16.6% (246,154/1,479,183) of outpatients and 29.2% (61,631/210,765) of outpatients and inpatients, respectively. The designated code gave much lower estimates (outpatients 0.49% (7,213/1,479,183), inpatients 2.6% (5,521/210,765)). Among 933,877 influenza patients, 17.0% (138,951/817,336) of outpatients and 24.6% (18,824/76,390) of inpatients

**Data Availability Statement:** Concerning data availability, the minimal data set is included in the Supporting information. All data in Supporting information can be used without restriction. This now includes the precise values used to build the long COVID symptom trend graphs (S1 Table) and the detailed statistical data obtained in the logistic

and Poisson regressions (S5 Table), from which the odds ratios and incidence rate ratios can be derived. As for raw data, CMS does not let us download (or distribute) any patient level data. The data stay on their machine, and we analyze it with software they provide on their machine. If researchers wish to access the raw data, they can contact the CMS Virtual Research Data Center https://resdac.org/cms-virtual-research-data-center-vrdc. However, data access requires the payment of a fee.

**Funding:** This research was supported by the Intramural Research Program of the National Library of Medicine (NLM), National Institutes of Health. All authors were full time employees of NLM. The funders had no role in study design, data collection and analysis, decision to publish, or preparation of the manuscript.

**Competing interests:** The authors have declared that no competing interests exist.

**Abbreviations:** CMS, Centers for Medicare and Medicaid Services; COVID-19, Coronavirus Disease 2019; ED, emergency department; FFS, fee-for-service; GEE, generalized estimating equation; HMO, health maintenance organization; ICD-10-CM, International Classification of Disease-10th Version-Clinical Modification; ME/CFS, myalgic encephalomyelitis/chronic fatigue syndrome; MERS-CoV, Middle East Respiratory Syndrome Coronavirus; PASC, post-acute sequelae SARS-CoV-2 infection; SARS-CoV-2, Severe Acute Respiratory Syndrome Coronavirus 2; SNF, skilled nursing facility; VRDC, Virtual Research Data Center; WHO, World Health Organization.

fit the long Flu definition. Long COVID patients had higher incidence of dyspnea, fatigue, palpitations, loss of taste/smell, and neurocognitive symptoms compared to long Flu. Long COVID outpatients were more likely to have any-cause hospitalization (31.9% (74,854/234,688) versus 26.8% (33,140/123,736), odds ratio 1.06 (95% CI 1.05 to 1.08, $p < 0.001$)), and more outpatient visits than long Flu outpatients (mean 2.9(SD 3.4) versus 2.5(SD 2.7) visits, incidence rate ratio 1.09 (95% CI 1.08 to 1.10, $p < 0.001$)). There were less ED visits in long COVID patients, probably because of reduction in ED usage during the pandemic. The main limitation of our study is that the diagnosis of long COVID in is not independently verified.

## Conclusions

Relying on specific long COVID diagnostic codes results in significant underreporting. We observed that about 30% of hospitalized COVID-19 patients developed long COVID. In a similar proportion of patients, long COVID-like symptoms (long Flu) can be observed after influenza, but there are notable differences in symptomatology between long COVID and long Flu. The impact of long COVID on healthcare utilization is higher than long Flu.

## Author summary

### Why was this study done?

- The quoted incidence of long COVID varies widely because of differences in definition and measurement method. Long COVID in the elderly is likely to be underreported because they are less likely to respond to surveys, and symptoms may be confused with other chronic diseases.

- Lingering ill health after infections is not limited to COVID-19. We postulate that some patients may fit the diagnostic criteria of long COVID after a bout of influenza. We call this condition "long Flu." Comparing and contrasting long COVID and long Flu may shed light on the understanding of long COVID, a disease still shrouded in mystery.

### What did the researchers do and find?

- We used the World Health Organization long COVID definition on 2 million Medicare patients with COVID-19 between April 2020 and June 2021. We applied the same definition to almost 900,000 influenza patients during the 2018 and 2019 Flu seasons to identify long Flu.

- Long COVID occurred in 16.6% of outpatients and 29.2% of inpatients. The corresponding rates for long Flu were 17% and 24.6%. Using only the designated long COVID code, the estimated rates would be 0.5% and 2.6%, way below the reported rates in most studies.

- Despite the similar overall incidence rates, long COVID patients suffered with notably different symptoms compared to long Flu patients and were also more likely to access inpatient and outpatient healthcare services.

**What do these findings mean?**

- The use of designated long COVID diagnostic codes alone is likely to result in gross underreporting of long COVID in this population.

- Long COVID is associated with greater healthcare utilization than long Flu, suggesting a bigger impact on individual health and well-being, as well as on healthcare expenditure.

## 1. Introduction

Most Coronavirus Disease 2019 (COVID-19) patients recover completely after an infection with the Severe Acute Respiratory Syndrome Coronavirus 2 (SARS-CoV-2 virus). However, a proportion suffer from persistent health issues after the acute phase of COVID-19 [1–7]. Various names have been used to describe this condition, including long COVID, long-haulers, long-term effects of COVID-19, post-COVID syndrome, chronic COVID syndrome, post-COVID conditions, and post-acute sequelae SARS-CoV-2 infection (PASC). We shall use long COVID in this report. Symptoms reported by long COVID patients range from fatigue, dyspnea, loss of smell to "brain fog." The incidence of long COVID varies widely between studies, the majority are between 10% and 30% [8–19]. According to one estimation, up to 23 million people in the United States may have developed long COVID as of February 2022 [20]. Another study estimated that at least 3 to 5 million US adults have activity-limiting long COVID [21].

While it is known that elderly patients are more prone to develop severe COVID-19, some studies have identified age as a risk factor also for long COVID [22,23]. So far, relatively few long COVID studies have focused on the elderly. Long COVID can be underreported in the elderly population because they may not be as troubled by, or ready to report, the symptoms as in younger people [24]. They may also be less likely to participate in internet-based research or respond to questionnaires. Moreover, the long COVID symptoms may be masked by or attributed to existing chronic diseases. One study finds that almost a third of COVID-19 patients over 65 years developed one or more new or persistent clinical sequelae [25]. Another study reports significant deterioration in quality of life and functional decline in elderly patients 6 months after COVID-19 [26]. More information is needed to understand the impact of long COVID on the elderly population.

While long COVID is sometimes considered a novel disease, it is hardly a totally unexpected phenomenon. Many viral or bacterial infections have been known to cause prolonged illnesses in a subset of patients [27]. Rheumatic fever following infection by *Streptococcus pyogenes* is a well-known example. Herpesviruses and enteroviruses are implicated in the cause of myalgic encephalomyelitis/chronic fatigue syndrome (ME/CFS), characterized by fatigue, musculoskeletal pain, and post-exertional malaise [28]. First reports of prolonged symptoms after contracting Russian influenza dated back to the 19th century [10,29]. More recently, the SARS-CoV-1 and the Middle East Respiratory Syndrome Coronavirus (MERS-CoV) have been associated with post-acute phase persistent symptoms that affected approximately one-third of patients [30].

There are a lot of similarities between COVID-19 and influenza. Both diseases are caused by easily transmissible single-stranded RNA viruses primarily affecting the respiratory tract, with significant systemic manifestations. Both diseases affect millions of patients every year,

presenting substantial medical and socioeconomic challenges. It is conceivable that, in some patients, a persistent state of ill health, similar to long COVID, can occur after influenza [31]. Given the similarities between COVID-19 and influenza, we postulate that there is considerable overlap in the symptomatology of the post-viral syndromes that they are associated with. This means that some patients would satisfy the diagnostic criteria for long COVID after an episode of influenza had the primary infection been COVID-19 instead. For lack of a better name, we shall call this condition "long Flu." Conceptually, long Flu patients can serve as an "influenza comparator group" for long COVID. We think that comparing long COVID and long Flu would bring new insights to the understanding of long COVID.

In this study, we develop a pragmatic algorithm to identify long COVID patients based on the clinical definition proposed by the World Health Organization (WHO). Furthermore, we apply the same algorithm to identify patients who may be suffering from long Flu in 2 previous influenza seasons (2018 and 2019) and compare them with long COVID patients in 3 aspects: incidence, symptomatology, and impact on healthcare utilization.

## 2. Materials and methods

### 2.1 Study population

The primary cohort of this study was all Medicare patients over 65 diagnosed to have COVID-19 between April 2020 and June 2021. A control cohort of non-COVID-19 patients was identified by 1 to 5 matching for the same period. An influenza comparator cohort was identified from 2 pre-pandemic influenza seasons in 2018 and 2019. Through the Virtual Research Data Center (VRDC) [32] of the Centers for Medicare and Medicaid Services (CMS), we accessed de-identified encounter data of all Medicare beneficiaries from 2016 to 2021. Medicare is the US federal government's health insurance program that primarily covers people 65 and older, and certain younger people with disabilities or kidney failure. Most individuals become eligible for Medicare when they reach 65 [33]. By one estimate, almost all (93%) of noninstitutionalized persons 65 and over, about 52 million in 2017, are covered by Medicare [34]. We focused our analysis only on Medicare beneficiaries aged ≥65, since younger Medicare beneficiaries are not representative of the general population aged <65 as they need qualifying disability conditions to enroll. To ensure we have sufficient data for symptom look-up (see below for method to identify long COVID and long Flu), we excluded patients (a) with less than 1 year of Medicare coverage, (b) with no encounters in a year prior to COVID-19 or influenza diagnosis, and (c) who were continuously enrolled in Medicare Advantage plans, mostly private health maintenance organization (HMO) plans that are not original Medicare fee-for-service (FFS) plans in the period between 1 year before and 12 weeks after the COVID-19 or influenza diagnosis. The last exclusion is necessary because Medicare claims data are potentially incomplete for patients enrolled in non-FFS plans. This study was declared not human subject research by the Office of Human Research Protection at the National Institutes of Health and by the CMS's Privacy Board. There was no prospective analysis protocol submitted before the commencement of this study.

### 2.2 Identifying long COVID and long Flu

**2.2.1 Long COVID.** We identified COVID-19 patients based on the International Classification of Disease-10th Version-Clinical Modification (ICD-10-CM) code *U07.1 COVID-19*, in either inpatient or outpatient claims between April 1, 2020, and June 30, 2021. We stopped at June 2021 to ensure that we have acquired complete data for long COVID analysis because of several months' lag for claim maturity. We separated COVID-19 patients into 2 mutually exclusive groups: outpatient and inpatient. For outpatients, the first COVID-19 diagnosis

must be an outpatient coding, and the patient must not be admitted to an inpatient facility (acute care hospital or skilled nursing facility (SNF)) for COVID-19 within 4 weeks of COVID-19 diagnosis. An SNF is an inpatient rehabilitation and medical treatment center that provides a wide range of medical care including physical therapy, intravenous therapy, injections, monitoring of vital signs, and medical equipment. For inpatients, (a) the first COVID-19 diagnosis must be an inpatient coding, or (b) the first COVID-19 diagnosis is an outpatient coding, but the patient must be admitted to an inpatient facility for COVID-19 within 4 weeks of the COVID-19 diagnosis.

For identification of long COVID, we tried 2 approaches. The first was based on the recommended ICD-10-CM code for long COVID, *B94.8 Sequelae of other specified infectious and parasitic diseases*, during our study period (**code-based definition**). B94.8 can potentially be used in non-COVID-19 infections, but significant use of this code in Medicare data occurred only after April 2020 (usage increased over 20-fold), making it reasonably specific for long COVID. Note that after our study, a new specific code for long COVID, *U09.9 Post COVID-19 condition*, became available from October 2021. During peer review of this paper, additional analysis was suggested to study the impact of the new long COVID code on the diagnosis of long COVID. This was done using additional data from September to December 2021, which had become available after our study was concluded. The second approach was based on a constellation of symptoms (**symptom-based definition**). We followed the WHO's clinical definition that was developed through a consensus process involving over 200 experts, researchers and patients [35]:

> "*Post COVID-19 condition occurs in individuals with a history of probable or confirmed SARS-CoV-2 infection, usually 3 months from the onset of COVID-19 with symptoms that last for at least 2 months and cannot be explained by an alternative diagnosis. Common symptoms include fatigue, shortness of breath, cognitive dysfunction but also others which generally have an impact on everyday functioning. Symptoms may be new onset, following initial recovery from an acute COVID-19 episode, or persist from the initial illness. Symptoms may also fluctuate or relapse over time.*"

We used ICD-10-CM codes to identify the 11 symptoms that at least 50% of the participants in the WHO's consensus building process thought were critical to include (see S1 Table for their incidence and temporal trend). Long COVID was defined as presence of any of the 11 symptoms unless they were excluded (see below for exclusion criteria). We also did a sensitivity analysis using only the top 3 symptoms (fatigue, shortness of breath, cognitive dysfunction) that reached 70% agreement in the WHO's consensus building. For outpatients, we looked for symptoms from 4 to 12 weeks after the COVID-19 diagnosis. For inpatients, we started looking for long COVID symptoms after they were discharged to their original place of residence, following the recommendation from Amenta and colleagues [36]. The observation period for inpatients was from 2 to 10 weeks after discharge to compensate for the median hospitalization period of 2 weeks.

We did not include SNF patients (9% of all COVID-19 patients) in the inpatient group because the proportion of COVID-19 patients admitted to SNF was unusually high (32.6%) compared to influenza patients in 2 previous Flu seasons (4.2%). More importantly, 88.6% of SNF COVID-19 patients were not discharged home but transferred to different inpatient facilities at the end of our study period. Two CMS policy changes in response to the pandemic may explain these phenomena: (a) waiving of the 3-day requirement of prior hospital stay for admission to SNF; and (b) the extension of SNF coverage for an additional 100 days [37]. Since our identification of long COVID for inpatients started after their discharge, most of SNF patients did not satisfy the inclusion criteria.

To satisfy the requirement that "[the symptom] cannot be explained by an alternative diagnosis," we tried 2 approaches:

1. **Exclusion by history**—the long COVID symptom must not be present from 2 weeks to 1 year before the COVID-19 diagnosis. We started the look-back period 2 weeks prior to COVID-19 diagnosis because COVID-19-related symptoms started to increase from 2 weeks before the COVID-19 diagnosis, indicating a lag in diagnosis reporting (see S1 Table). Note that this exclusion only applied to individual symptoms, not to the patient as a whole. For example, if a patient had dyspnea 6 weeks after COVID-19 diagnosis, but the patient also complained of dyspnea 6 months before COVID-19 diagnosis, then dyspnea would not be counted as a long COVID symptom. However, the patient was not excluded and might still be identified as long COVID due to other symptoms.

2. **Exclusion by history and comorbidities**—in addition to 1 above, we excluded symptoms that could be explained by a known comorbidity, for example, dyspnea excluded in the presence of chronic obstructive pulmonary disease (see S2 Table for excluded symptoms). We used the chronic condition onset dates in the Medicare database to identify comorbidities [38].

To estimate the false positive rate of the symptom-based definition, we matched each outpatient COVID-19 case to 5 controls (never had the code *U07.1 COVID-19* or *Z86.16 Personal history of COVID-19*) on age in years, race, sex, dual eligibility status (a surrogate for income), geographic region, and Charlson comorbidity index [39] (within a range of +/− 2). We applied the same symptom-based definition on controls to assess the proportion of patients who would be falsely identified as long COVID.

**2.2.2 Long Flu.**   Since the incidence of influenza diminished significantly during the COVID pandemic, we used data from 2 pre-pandemic Flu seasons, October 2017 to May 2018 (2018 season) and October 2018 to May 2019 (2019 season) to estimate the incidence of the postulated long Flu. We identified influenza patients using the ICD-10-CM codes J09, J10, and J11. Similar to COVID-19, we separated influenza patients into outpatient and inpatient groups. We used the same list of long COVID symptoms and symptom exclusion criteria to identify long Flu. The observation period was the same as for long COVID, i.e., from 4 to 12 weeks after influenza diagnosis for outpatients, and 2 to 10 weeks after discharge for inpatients. We excluded all SNF influenza inpatients (0.5% of total) from the long Flu cohort to be comparable with the long COVID cohort.

## 2.3 Comparing long COVID and long Flu

We compared the incidence and the distribution of symptoms between long COVID and long Flu patients. To estimate the impact of long COVID or long Flu on healthcare utilization, we analyzed 4 outcomes: (a) hospitalization (any cause); (b) hospitalization (due to any long COVID symptom); (c) emergency department (ED) visit (due to any long COVID symptom); and (d) number of outpatient (excluding ED) encounters (due to any long COVID symptom). We ran analyses of each of 4 outcomes separately for outpatients and hospitalized patients. For outpatients, we observed the outcomes for the period 4 to 12 weeks post-COVID-19 or influenza diagnosis. For inpatients, the observation period was 2 to 10 weeks post-discharge.

## 2.4 Statistical analysis

We included all patients with COVID-19 or influenza, including those diagnosed with both. When we had to compare COVID-19 and influenza patients statistically, we excluded patients who had both conditions to ensure independence between groups. To test the difference in

incidence of each specific symptom between long COVID and long Flu patients, we first implemented a two-by-two contingency chi-squared test [40] and used Hochberg method to control for familywise error rate from multiple hypotheses testing by decreasing the number of false positives [41]. We then used a multiple logistic regression model [42] to compare them further controlling for age, sex, race, region, dual eligibility, and Charlson comorbidity index and reported the adjusted odds ratios. We did not have to deal with missing data as demographics and socioeconomic data were always present.

To test the difference in the first 3 outcomes of healthcare utilization (hospitalization any cause, hospitalization with long COVID symptoms, ED visit with long COVID symptoms), we implemented a generalized linear model with logit link adjusting for all available patient characteristics (age, sex, race, geographical region, dual eligibility status, history of any hospitalization in prior year, and 55 chronic conditions) as covariates. We adjusted for these covariates because demographics, comorbidities, and socioeconomic factors are known to affect healthcare utilization. To compare the number of outpatient visits with long COVID symptoms, we used a generalized linear model with a log link function (i.e., Poisson regression or log-linear regression analysis) with the same set of covariates as adjusters. We used the generalized estimating equations (GEEs) method to account for overdispersion in the Poisson regression model.

The primary goal of this study is to estimate the incidence of long COVID in the elderly by various approaches (code-based and symptom-based). Furthermore, we aim to compare the incidence, symptomatology, and healthcare utilization of long COVID with the hypothetical condition of long Flu. This study is reported as per the Strengthening the Reporting of Observational Studies in Epidemiology (STROBE) guideline (S1 Checklist).

## 3. Results

### 3.1 Patient characteristics

We began with 2,434,154 COVID-19, 1,033,515 influenza, and 8,755,799 control Medicare beneficiaries aged ≥65. After applying exclusion criteria discussed in 2.1., we were left with 2,071,532 COVID-19 patients of whom 1,479,183 (71.4%) were outpatients (Fig 1). We combined data from the 2 Flu seasons (2018 and 2019) because they were quite similar in terms of patients' health status and demographics (see S3 Table). There were 933,877 influenza patients of whom 817,336 (87.5%) were outpatients. All cases and controls were followed-up for 2 months to identify long COVID or long Flu, making a total follow-up time of 1.5 million patient years.

Table 1 shows the characteristics of the COVID-19 and influenza patients. Note that in comparing 2 groups, 91,796 COVID-19 patients who also had influenza in previous 2 Flu seasons were excluded (3% of total). Among the hospitalized patients, influenza patients were older and sicker (higher prior hospitalization rate and Charlson comorbidity index) than COVID-19 patients, but the difference was much smaller among outpatients.

Each COVID-19 outpatient was matched to 5 non-COVID-19 patients. Overall, there were 6,286,633 controls with unmatched rate (COVID-19 patients unable to be matched) of 1.1%. The cases and controls were generally quite well matched in terms of demographics and Charlson comorbidity index. Due to the large sample size, most differences between cases and controls were statistically significant, but the absolute standardized difference <0.25 indicated good balance between them (S4 Table) [43].

### 3.2 Incidence of long COVID and long Flu

Among the hospitalized COVID-19 and influenza patients, 52.8% (210,765/399,124) and 68.4% (76,390/111,721), respectively, were discharged to their original place of residence by

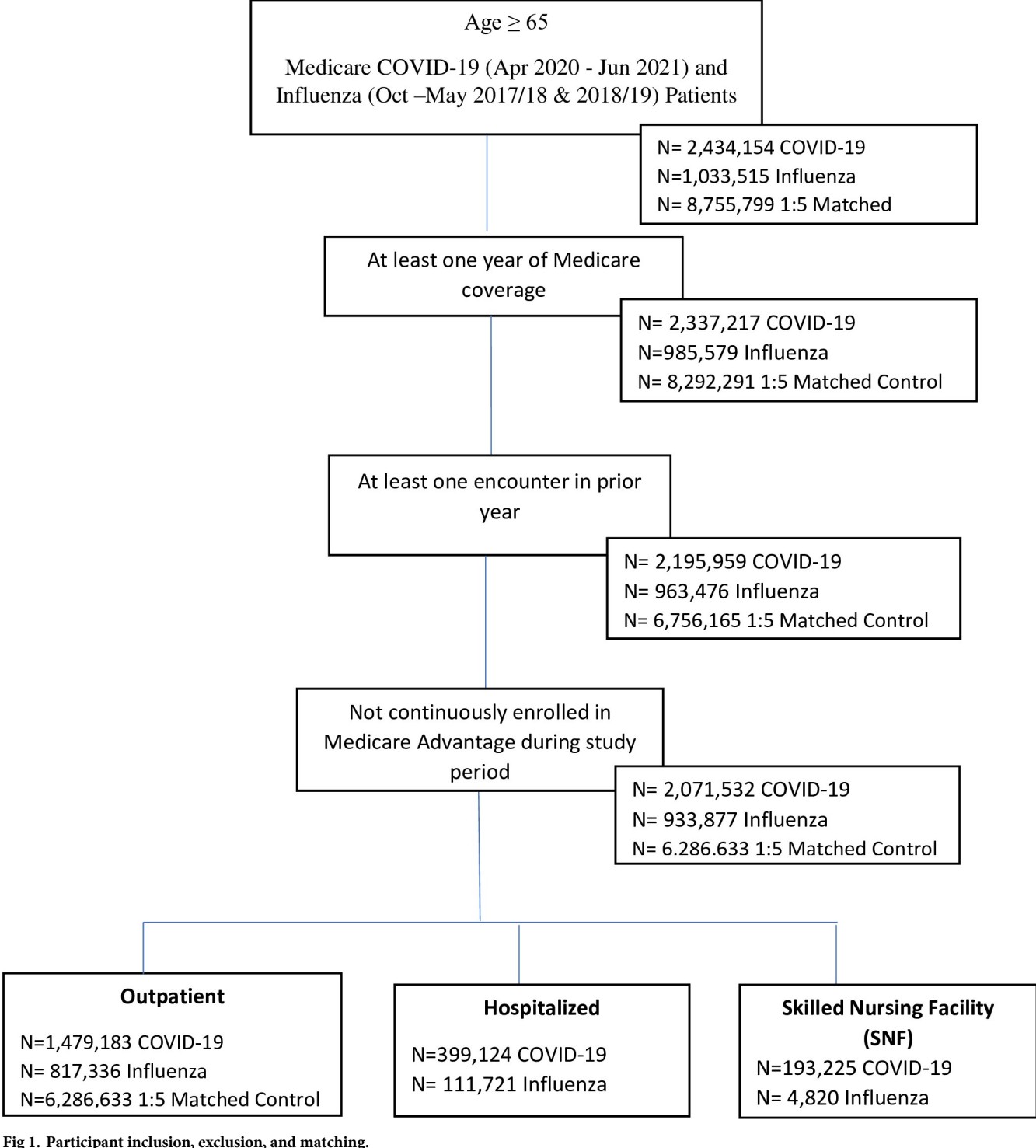

**Fig 1. Participant inclusion, exclusion, and matching.**

the end of our study period. These patients were used to estimate the incidence of long COVID and long Flu in inpatients. Based on the ICD-10-CM diagnosis code of B94.8 (i.e., the code-based definition), only 0.49% of outpatients and 2.6% of hospitalized patients were identified to develop long COVID (Table 2), way lower than most published reports. Using the

**Table 1. Characteristics of COVID-19 and influenza patients (CCI, Charlson comorbidity index, IQR, interquartile range; LIS, low-income subsidy; SD, standard deviation).**

| | Outpatient | | | Inpatient (hospitalized) | | |
|---|---|---|---|---|---|---|
| | COVID-19 | Influenza[a] | P value[b] | COVID-19 | Influenza[a] | P value[b] |
| N (%) | 1,479,183 (100) | 817,336 (100) | | 399,124 (100) | 111,721 (100) | |
| Any prior hospitalization[c] | 349,688(23.6) | 174,024(21.3) | <0.001 | 121,372(30.4) | 42,868(38.4) | <0.001 |
| CCI, mean ± SD | 1.9±2.3 | 1.9±2.2 | <0.001 | 2.6±2.6 | 3.0±2.6 | <0.001 |
| Age, median (IQR) | 75.0(70.0–82.0) | 74.0(70.0–81.0) | <0.001 | 78.0(72.0–85.0) | 81.0(74.0–88.0) | <0.001 |
| 65–69 | 313,613(21.2) | 185,573(22.7) | <0.001 | 58,643(14.7) | 11,826(10.6) | <0.001 |
| 70–74 | 389,872(26.4) | 225,954(27.6) | <0.001 | 86,543(21.7) | 18,726(16.8) | <0.001 |
| 75–79 | 277,075(18.7) | 160,188(19.6) | <0.001 | 79,671(20.0) | 20,232(18.1) | <0.001 |
| 80–84 | 205,435(13.9) | 110,568(13.5) | <0.001 | 70,961(17.8) | 20,733(18.6) | <0.001 |
| 85+ | 293,188(19.8) | 135,053(16.5) | <0.001 | 103,306(25.9) | 40,204(36.0) | <0.001 |
| Female | 853,157(57.7) | 491,113(60.1) | <0.001 | 201,712(50.5) | 65,020(58.2) | <0.001 |
| Race: White | 1,165,494(78.8) | 664,540(81.3) | <0.001 | 296,000(74.2) | 92,339(82.7) | <0.001 |
| Black | 117,273(7.9) | 55,488(6.8) | <0.001 | 46,349(11.6) | 8,637(7.7) | <0.001 |
| Hispanic | 111,488(7.5) | 50,594(6.2) | <0.001 | 34,782(8.7) | 6,176(5.5) | <0.001 |
| Asian | 37,698(2.5) | 24,401(3.0) | <0.001 | 9,870(2.5) | 2,448(2.2) | <0.001 |
| Other | 47,230(3.2) | 22,313(2.7) | <0.001 | 12,123(3.0) | 2,121(1.9) | <0.001 |
| Region: Northeast | 322,838(21.8) | 131,762(16.1) | <0.001 | 76,301(19.1) | 22,251(19.9) | <0.001 |
| Midwest | 309,689(20.9) | 160,077(19.6) | <0.001 | 97,444(24.4) | 30,757(27.5) | <0.001 |
| South | 566,699(38.3) | 384,311(47.0) | <0.001 | 164,849(41.3) | 38,622(34.6) | <0.001 |
| West | 265,878(18.0) | 129,759(15.9) | <0.001 | 57,326(14.4) | 18,873(16.9) | <0.001 |
| Income: Ever Dual | 367,459(24.8) | 156,074(19.1) | <0.001 | 108,094(27.1) | 28,562(25.6) | <0.001 |
| Non-Dual LIS | 21,666(1.5) | 14,397(1.8) | <0.001 | 8,806(2.2) | 2,466(2.2) | <0.001 |
| Non-Dual Non-LIS | 1,090,058(73.7) | 646,865(79.1) | <0.001 | 282,224(70.7) | 80,693(72.2) | <0.001 |

[a]Combined 2018 and 2019 Flu seasons.

[b]Excluding 91,796 patients with both COVID-19 and influenza.

[c]Any prior hospitalization within 1 year of COVID-19 or influenza diagnosis.

symptom-based definition, the estimated incidence of long COVID was closest to other studies when we applied the definition of "any of 11 symptoms with history exclusion only." By this definition, the incidence of long COVID was 16.6% and 29.2% in outpatients and hospitalized patients, respectively. If we added the comorbidity exclusion to this definition, the rates would

**Table 2. Incidence of long COVID and long Flu by various definitions.**

| Long COVID definition | Outpatient | | Inpatient (hospitalized) | | Control |
|---|---|---|---|---|---|
| | COVID-19 | Influenza | COVID-19 | Influenza | Matched with COVID-19 outpatient |
| N(%) | 1,479,183(100) | 817,336(100) | 210,765(100) | 76,390(100) | 6,286,633(100) |
| Code-based[a] | 7,213(0.49) | 39(0.005) | 5,521(2.6) | 11(0.01) | 264 (0.004) |
| Symptom-based | | | | | |
| Any of 11 symptoms with history exclusion only | 246,154(16.6) | 138,951(17.0) | 61,631(29.2) | 18,824(24.6) | 658,229(10.5) |
| Any of 11 symptoms with history and comorbidity exclusion | 84,774(5.7) | 50,616(6.2) | 18,486(8.8) | 4,804(6.3) | 220,751(3.5) |
| Any of 3 main symptoms with history exclusion only | 142,175(9.6) | 70,680(8.7) | 42,690(20.3) | 10,907(14.3) | 357,915(5.7) |
| Any of 3 main symptoms with history and comorbidity exclusion | 54,754(3.7) | 30,053(3.7) | 13,531(6.4) | 3,302(4.3) | 148,251(2.4) |

[a]Using code B94.8.

drop to 5.7% (outpatient) and 8.8% (inpatient). We shall use the "any of 11 symptoms with history exclusion only" definition as our main result in subsequent discussion.

### 3.3 Difference in symptomatology of long COVID and long Flu

After excluding 91,796 patients with both COVID-19 and influenza (3% of all COVID-19 and influenza patients), we had 293,172 long COVID patients (outpatient 234,688, inpatient 58,484) and 140,697 long Flu patients (outpatient 123,736, inpatient 16,961) (Table 3). Rates of dyspnea, fatigue, palpitations, and loss of taste/smell were significantly higher in long COVID than long Flu, among both outpatients and hospitalized patients. In contrast, cough, chest pain, headache, and muscle/joint pain were more frequent in long Flu in both outpatient and inpatient groups. The incidence of memory problem, cognitive impairment, and sleep disturbance were significantly higher in long COVID among outpatients only.

### 3.4 Difference in healthcare utilization in long COVID and long Flu

For both outpatients and hospitalized patients, long COVID was associated with significantly higher chance of any hospitalizations and more outpatient visits than long Flu (Table 4). The

**Table 3. Symptoms in long COVID and long Flu patients (CI, confidence interval; OR, odds ratio of developing each specific symptom for long COVID compared to long Flu).**

| Symptoms | Long COVID (%) | Long Flu (%) | P value | | OR (95% CI) | |
|---|---|---|---|---|---|---|
| | | | Unadjusted | Adjusted[a] | Unadjusted | Adjusted[b] |
| **Outpatient** | N = 234,688(100) | N = 123,736(100) | | | | |
| dyspnea | 66,562(28) | 29,689(24) | <0.001 | <0.001 | 1.25(1.23,1.27) | 1.25(1.23,1.27) |
| fatigue/malaise/weakness | 76,881(33) | 36,547(30) | <0.001 | <0.001 | 1.16(1.15,1.18) | 1.15(1.13,1.16) |
| cough | 45,776(20) | 32,192(26) | <0.001 | <0.001 | 0.69(0.68,0.70) | 0.69(0.67,0.70) |
| chest pain | 41,697(18) | 22,538(18) | 0.001 | 0.002 | 0.97(0.95,0.99) | 0.97(0.95,0.99) |
| palpitations | 31,120(13) | 13,940(11) | <0.001 | <0.001 | 1.20(1.18,1.23) | 1.20(1.18,1.23) |
| headache | 17,950(8) | 10,637(9) | <0.001 | <0.001 | 0.88(0.86,0.90) | 0.89(0.87,0.91) |
| muscle/joint pain | 14,097(6) | 8,332(7) | <0.001 | <0.001 | 0.89(0.86,0.91) | 0.91(0.88,0.93) |
| memory problem | 8,261(4) | 4,097(3) | 0.001 | 0.002 | 1.07(1.03,1.11) | 1.09(1.05,1.13) |
| cognitive impairment | 6,698(3) | 2,671(2) | <0.001 | <0.001 | 1.33(1.27,1.39) | 1.29(1.23,1.35) |
| sleep disturbance | 2,360(1) | 1,148(0.9) | 0.024 | 0.024 | 1.08(1.01,1.16) | 1.10(1.02,1.18) |
| loss of taste/smell | 1,537(0.7) | 414(0.3) | <0.001 | <0.001 | 1.96(1.76,2.19) | 2.02(1.81,2.26) |
| **Inpatient (hospitalized)** | N = 58,484(100) | N = 16,961(100) | | | | |
| dyspnea | 24,915(43) | 5,186(31) | <0.001 | <0.001 | 1.69(1.62,1.75) | 1.51(1.45,1.57) |
| fatigue/malaise/weakness | 20,500(35) | 5,424(32) | <0.001 | <0.001 | 1.15(1.11,1.19) | 1.25(1.21,1.30) |
| cough | 11,425(20) | 4,247(25) | <0.001 | <0.001 | 0.73(0.70,0.76) | 0.69(0.66,0.72) |
| chest pain | 9,773(17) | 3,065(18) | <0.001 | <0.001 | 0.91(0.87,0.95) | 0.91(0.87,0.95) |
| palpitations | 8,467(14) | 2,054(12) | <0.001 | <0.001 | 1.23(1.17,1.29) | 1.22(1.15,1.28) |
| headache | 3,026(5) | 1,288(8) | <0.001 | <0.001 | 0.66(0.62,0.71) | 0.71(0.66,0.76) |
| muscle/joint pain | 2,103(4) | 715(4) | <0.001 | <0.001 | 0.85(0.78,0.92) | 0.86(0.79,0.94) |
| memory problem | 1,849(3) | 648(4) | <0.001 | 0.034 | 0.82(0.75,0.90) | 0.91(0.83,1.00) |
| cognitive impairment | 1,066(2) | 401(2) | <0.001 | <0.001 | 0.77(0.68,0.86) | 0.89(0.79,1.01) |
| sleep disturbance | 508(0.9) | 137(0.8) | 0.448 | 0.448 | 1.08(0.89,1.30) | 1.09(0.90,1.33) |
| loss of taste/smell | 288(0.5) | 42(0.3) | <0.001 | <0.001 | 1.99(1.44,2.76) | 2.04(1.47,2.83) |

[a]Adjusted for multiple hypothesis testing from chi-squared test.

[b]Adjusted for age, sex, race, region, dual eligibility, and Charlson comorbidity index.

**Table 4. Healthcare utilization of long COVID and long Flu patients (CI, confidence interval; ED, emergency department; IRR, incidence rate ratio; OR, odds ratio; SD, standard deviation).**

| | Healthcare utilization | | | |
| --- | --- | --- | --- | --- |
| | Any hospitalization (%) | Symptom-specific[a] hospitalization (%) | Symptom-specific[a] ED visit (%) | Number of symptom-specific[a] outpatient visits, mean (SD) |
| Outpatient | | | | |
| Long COVID, $N$ = 234,688 | 74,854(31.9) | 36,721(15.6) | 25,131(10.7) | 2.9(3.4) |
| Long Flu, $N$ = 123,736 | 33,140(26.8) | 16,065(13.0) | 15,566 (12.6) | 2.5(2.7) |
| Inpatient (hospitalized) | | | | |
| Long COVID, $N$ = 58,484 | 20,182(34.5) | 8,286(14.2) | 7,816(13.4) | 2.9(3.2) |
| Long Flu, $N$ = 16,961 | 6,245(36.8) | 2,593(15.3) | 2,749(16.2) | 2.8(3.0) |
| Comparative risk in long COVID compared to long Flu | OR (95%CI, *p*-value) | OR (95%CI, *p*-value) | OR (95%CI, *p*-value) | IRR (95%CI, *p*-value) |
| Unadjusted | | | | |
| Outpatient | 1.28(1.26,1.30, <0.001) | 1.24(1.22,1.27, <0.001) | 0.83(0.82,0.85, <0.001) | 1.16(1.15,1.16, <0.001) |
| Inpatient (hospitalized) | 0.90(0.87,0.94, <0.001) | 0.91(0.87,0.96, <0.001) | 0.80(0.76,0.84, <0.001) | 1.02(1.01,1.04, 0.009) |
| Adjusted[b] | | | | |
| Outpatient | 1.06(1.05,1.08, <0.001) | 1.00(0.98,1.02, 0.979) | 0.80(0.78,0.82, <0.001) | 1.09(1.08,1.10, <0.001) |
| Inpatient (hospitalized) | 1.24(1.19,1.29, <0.001) | 1.20(1.14,1.27, <0.001) | 0.90(0.86,0.95, <0.001) | 1.09(1.07,1.11, <0.001) |

[a]With any of the 11 symptoms of long COVID.

[b]Adjusted for age, sex, race, geographical region, dual eligibility status, history of any hospitalization in prior year, and chronic comorbidities.

difference is especially notable among hospitalized patients, even though hospitalized influenza patients, at the baseline, were sicker and older than hospitalized COVID-19 patients, which should normally translate into more healthcare utilization. In contrast, the likelihood of an ED visit was significantly higher among long Flu patients.

## 4. Discussion

Using Medicare data and a symptom-based definition, we estimate that long COVID happens in 16.6% of outpatients and 29.2% of hospitalized patients that are over 65. We find notable differences between the pattern of symptoms of long COVID and the hypothetical condition of long Flu. Furthermore, long COVID patients utilize more healthcare services than long Flu.

One major hurdle in long COVID research is the difficulty in identifying long COVID cases. Due to the wide range of definitions, the results of studies often cannot be compared or generalized [12]. The development of a consensus on clinical definition by WHO partially fills this gap [35]. Our study has developed a method to operationalize this clinical definition. The WHO definition depends on the presence of a constellation of symptoms but requires that the symptoms do not have an alternative explanation. Identifying such explanations can be done in observational studies. We compared 2 approaches—exclusion by history (the symptom did not occur within the previous year) and exclusion by history and comorbidities. The use of exclusion by history alone yields estimates of long COVID of 16.6% in outpatients and 29.2% in hospitalized patients, which are close to many published results. A recent large meta-analysis covering 194 studies shows that on average, at least 45% of COVID-19 survivors (nonhospitalized patients 34.5%, hospitalized patients 52.6%) continue to experience at least one unresolved symptom [44]. This means that our results are likely to be an underestimation.

One potential problem of exclusion of individual symptoms based on history is that common symptoms are more likely to be excluded, even if they are indeed related to long COVID. In our study, the incidence of cognitive impairment and memory problem among long COVID patients is lower than that reported in the literature [44]. Since these problems are generally more common among elderly patients, they are more likely to be excluded by coincidence. Another factor that can be at play is that the threshold of seeking help for these problems may be higher in elderly patients. Adding the exclusion based on comorbidities cuts the estimates significantly to 5.7% for outpatients and 8.8% for hospitalized patients. We speculate that the comorbidity exclusion is probably too strict because some comorbidities are very prevalent in our study population of elderly Medicare beneficiaries. For example, high prevalence of heart failure (43%), chronic obstructive pulmonary disease (41%), and fibromyalgia chronic pain and fatigue (54%) would lead to exclusion of cough, dyspnea, and fatigue.

A recent study shows promise in the use of machine learning to identify long COVID [45]. However, the model in Pfaff and colleagues was only built on a very small, 0.6%, subset of 97,995 COVID-19 patients attending a long COVID clinic, highlighting the difficulty of identifying *all* patients suffering from long COVID. For machine learning methods to be effective, a large number of cases along with high-quality data is required. Achieving this goal can be difficult because long COVID tends to be underreported and undercoded [46]. In our study, code-based identification of long COVID yields an estimation of only 0.49% in outpatients and 2.6% in hospitalized patients, way below published results. A specific long COVID code, U09.9, was delivered in October 2021; however, healthcare providers were slow to take advantage of this code [47]. Based on additional data from September to December 2021, which became available after our study concluded, we found that the new long COVID code (U09.9) was used more often than the old one (B94.8), whose usage dropped off significantly. Using the new code in our code-based definition, 2.0% (11,424/573,965) of outpatients and 9.2% (6,253/68,030) of hospitalized patients developed long COVID. This is still considerably lower than most reports in the literature. Researchers should be aware of the potential underreporting if they rely solely on specific codes to identify long COVID.

We postulate the existence of "long Flu" based on reports of post-infectious sequelae after influenza [29]. Long COVID has attracted special attention because of the pandemic, but the possible occurrence of prolonged symptoms from influenza should not be overlooked. Long COVID is still a poorly understood disease. Comparing and contrasting long COVID with long Flu may offer new insights into its pathogenesis and treatment. The pathogenesis of long COVID is likely to be complex and more than one mechanism may be implicated in some clinical manifestations [28,48]. Evidence suggests that prolonged inflammation probably plays a key role. In addition, it is known that, like other coronaviruses, SARS-CoV-2 can invade the blood–brain barrier and access the central nervous system through peripheral or olfactory neurons [49,50]. This could explain the greater incidence of psychoneurological symptoms (for example, cognitive impairment, loss of taste or smell, memory problem, and sleep disturbance) in long COVID compared to long Flu in our study. Another special feature of COVID-19 is the high incidence of thromboembolism, probably as a result of endothelial injury and heightened inflammation, which can lead to organ or tissue injury [51,52]. Investigators have found high incidence of significant radiological and functional abnormalities indicative of lung parenchymal and small airway disease after the acute phase of COVID-19, which could give rise to dyspnea and easy fatigue [53].

Based on our estimation, the incidence of long COVID and long Flu is comparable among outpatients (16.6% versus 17.0%) and slightly higher for long COVID in inpatients (29.2% versus 24.6%). But incidence alone does not tell the whole story about the impact of the 2 diseases. Our model on healthcare utilization shows that long COVID patients are more likely to seek

outpatient care and be hospitalized, after controlling for demographics, socioeconomic factors, and comorbidities. This suggests that long COVID is a more serious illness than long Flu and has greater societal impact. One unexpected finding is the reduced ED visits in long COVID patients. This is probably an anomaly and could be explained by the fact that overall, ED visits shrunk during the pandemic, possibly because patients fearing exposure to COVID-19 avoided the ED for conditions for which they otherwise would have sought emergency care. Long Flu data came from 2017 to 2019, and we observed that the overall usage of ED dropped by 18% in 2020 compared with 2017 to 2019 among Medicare senior beneficiaries.

The primary strength of our study is the sample size. We have 2 million COVID-19 patients over 65, which is considerably more than most published studies. We recognize the following limitations. Based on our exclusion criteria, 15% of COVID-19 patients, 10% of influenza patients, and 28% of matched controls were excluded, which may affect the generalizability of our findings. Not all COVID-19 diagnoses are captured in Medicare claims data. Our previous study showed that up to one-third of COVID-19 cases could be missed [54]. Medicare claims-based data may miss services or treatment paid for by private insurance or other means. The code-based definition of long COVID is based on the recommended code available for the study period, which may not be specific for long COVID. The symptom-based definition relies on the symptoms reported as "diagnosis" at the healthcare encounter and may not be sensitive because providers may not routinely code all symptoms. Using claims data, we cannot easily ascertain the duration of the symptoms as stipulated in the WHO definition. There is no independent confirmation that patients identified by our method are indeed suffering from long COVID. However, we can venture some estimation of our error rates. False positive rate can be estimated by the positive rate in controls (10.5%). Among the 12,734 patients specifically coded as long COVID (code-based definition), 8,239 (64.7%) patients were identified as long COVID by the symptom-based definition, so the false negative rate can be estimated to be about 35%. If we adjust our results by these estimated error rates, the incidence of long COVID would be 9% for outpatients and 28% for inpatients, still not far from other studies. Our observation period ends at 12 weeks after the COVID-19 or influenza diagnosis. Some patients may present with long COVID or long Flu after that period. Among the inpatients, we exclude SNF patients because a significant proportion of them are not discharged within the study period. We assume that long Flu is similar to long COVID and use the same symptomatic definition. There may be symptoms in post-influenza syndrome that are not common in long COVID, and patients with those symptoms may not be identified as long Flu in our study.

Based on a constellation of symptoms identified in the WHO's consensus definition, we estimate that long COVID occurs in 16.6% and 29.2% of elderly COVID-19 outpatients and inpatients, respectively. The corresponding incidence for long Flu, identified by the same constellation of symptoms for 2 pre-pandemic influenza seasons, is about the same (17% and 24.6%). Long COVID patients have significantly higher incidence of dyspnea, fatigue, palpitations, loss of taste or smell, and neurocognitive symptoms. Compared to long Flu, patients with long COVID are hospitalized more often and have more outpatient visits, suggesting that it is a more serious illness and has higher societal impact.

## Supporting information

**S1 Checklist. Strengthening the Reporting of Observational Studies in Epidemiology (STROBE) guideline checklist.**
(DOCX)

**S1 Table. Long COVID symptoms and temporal trends.**
(XLSX)

**S2 Table. Exclusion of symptoms by comorbidities.**
(DOCX)

**S3 Table. Patient characteristics in the 2018 and 2019 Flu seasons.**
(DOCX)

**S4 Table. COVID-19 outpatient cases and controls.**
(DOCX)

**S5 Table. Detailed statistical data of the logistic and Poisson regression models for health-care utilization.**
(XLSX)

## Author Contributions

**Conceptualization:** Kin Wah Fung, Fitsum Baye, Seo H. Baik, Clement J. McDonald.

**Data curation:** Fitsum Baye, Seo H. Baik, Zhaonian Zheng.

**Formal analysis:** Kin Wah Fung, Fitsum Baye, Seo H. Baik, Zhaonian Zheng.

**Investigation:** Kin Wah Fung, Clement J. McDonald.

**Methodology:** Kin Wah Fung, Fitsum Baye, Seo H. Baik, Clement J. McDonald.

**Resources:** Clement J. McDonald.

**Supervision:** Clement J. McDonald.

**Writing – original draft:** Kin Wah Fung, Fitsum Baye, Seo H. Baik, Clement J. McDonald.

**Writing – review & editing:** Kin Wah Fung, Fitsum Baye, Seo H. Baik, Zhaonian Zheng, Clement J. McDonald.

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
