## [Editor Report · Decision Letter 0]

15 Sep 2022

Dear Dr Fung, 

Thank you for submitting your manuscript entitled "Long COVID in Elderly Patients: An Epidemiologic Exploration Using Medicare Data" for consideration by PLOS Medicine.

Your manuscript has now been evaluated by the PLOS Medicine editorial staff as well as by an academic editor with relevant expertise and I am writing to let you know that we would like to send your submission out for external peer review.

Please re-submit your manuscript within two working days, i.e. by Sep 19 2022 11:59PM.

Sincerely,

Philippa Dodd, MBBS MRCP PhD

Senior Editor

PLOS Medicine

---

## [Decision Letter · Decision Letter 1]

3 Jan 2023

Dear Dr. Fung,

Thank you very much for submitting your manuscript "Long COVID in Elderly Patients: An Epidemiologic Exploration Using Medicare Data" (PMEDICINE-D-22-03023R1) for consideration at PLOS Medicine. 

[LINK]

In light of these reviews, I am afraid that we will not be able to accept the manuscript for publication in the journal in its current form, but we would like to consider a revised version that addresses the reviewers' and editors' comments. Obviously we cannot make any decision about publication until we have seen the revised manuscript and your response, and we plan to seek re-review by one or more of the reviewers. 

We expect to receive your revised manuscript by Jan 24 2023 11:59PM. Please email us (plosmedicine@plos.org) if you have any questions or concerns.

We look forward to receiving your revised manuscript. 

Sincerely,

Philippa Dodd, MBBS MRCP PhD

PLOS Medicine

plosmedicine.org

GENERAL

Please respond to all editor and reviewer comments detailed below in full.

Please insert line numbers throughout the manuscript starting with 1 on page 2 at “Abstract” and in continuous sequence thereafter

Please ensure that the study is reported according to the STROBE guideline (we note the inclusion of a CONSORT diagram which is has greater relevance to RCTs) and include the completed STROBE checklist as Supporting Information. Please add the following statement, or similar, to the Methods: "This study is reported as per the Strengthening the Reporting of Observational Studies in Epidemiology (STROBE) guideline (S1 Checklist)."

* We agree with the reviewers and the academic editor that further details regarding Medicare eligibility and the study population would be helpful

** In light of reviewer #1 comments (below) regarding the use of the term “long-flu”, please consider how you frame the use of this term and in the context of this study and its aims

COMMENTS FROM THE ACADEMIC EDITOR

I think this is an interesting paper. I would recommend to ask the authors for a major revision to accommodate the comments of the reviewers. They should describe in more detail who is eligible for Medicare and what the characteristics are of the Medicare population, as this is not common knowledge outside the USA.

DATA AVAILABILITY STATEMENT

Thank you for including a statement regarding your data availability. The Data Availability Statement (DAS) requires revision.

PLOS Medicine requires that the de-identified data underlying the specific results in a published article be made available, without restrictions on access, in a public repository or as Supporting Information at the time of article publication, provided it is legal and ethical to do so. Please see the policy at 

http://journals.plos.org/plosmedicine/s/data-availability

and FAQs at 

http://journals.plos.org/plosmedicine/s/data-availability#loc-faqs-for-data-policy

For each data source used in your study: 

ABSTRACT

Please structure your abstract using the PLOS Medicine headings (Background, Methods and Findings, Conclusions).

Abstract Background: please remove “and importance”. The final sentence of the background section should clearly state the study question.

Abstract Methods and Findings:

Please ensure that all numbers presented in the abstract are present and identical to numbers presented in the main manuscript text.

Please include additional details regarding the medicare population and the study setting - rural, urban, pan-US or restricted to certain regions, total number of participants included in total and in each of the long covid and long-flu cohorts. 

You state “Long Flu was identified by the same symptom-based definition” – were these the same symptoms as used to define long-covid or a different set devised around the same concept? Please clarify. Please include further details of the long-flu cohort – over which years were cases of flu identified, was the time frame for identification of long-flu the same as for covid ? And so on...

Please clearly define the main outcome measures.

Please include any important dependent variables that are adjusted for in the analyses.

Please ensure percentages are quantified with numerators and denominators 

Please include the actual amounts and/or absolute risk(s) of relevant outcomes (including NNT or NNH where appropriate), not just relative risks or correlation coefficients. (example for absolute risks: PMID: 28399126). 

Please quantify the main results with 95% CIs and p values.

In the last sentence of the Abstract Methods and Findings section, please describe the main limitation(s) of the study's methodology.

Abstract Conclusions:

Please address the study implications without overreaching what can be concluded from the data; the phrase "In this study, we observed ..." may be useful.

Please interpret the study based on the results presented in the abstract, emphasizing what is new without overstating your conclusions.

Please avoid vague statements such as "these results have major implications for policy/clinical care". Mention only specific implications substantiated by the results.

Please avoid assertions of primacy ("We report for the first time....")

AUTHOR SUMMARY

METHODS and RESULTS

We agree with the reviewer (please see below) and with the academic editor (see above) that some additional detail regarding Medicare, its coverage and the population included would be helpful to the reader, please revise accordingly

Did your study have a prospective protocol or analysis plan? Please state this (either way) early in the Methods section.

For all observational studies, in the manuscript text, please indicate: 

(1) the specific hypotheses you intended to test, 

(2) the analytical methods by which you planned to test them, 

(3) the analyses you actually performed, and 

(4) when reported analyses differ from those that were planned, transparent explanations for differences that affect the reliability of the study's results. If a reported analysis was performed based on an interesting but unanticipated pattern in the data, please be clear that the analysis was data-driven.

Section 3.4, page 12 – the second half of this paragraph, sentence beginning “This is probably an anomaly…” should be moved to the relevant part of the discussion, suggest paragraph 2 of page 14 where healthcare utilization estimates are mentioned.

TABLES and FIGURES

Please ensure that all figures and tables have appropriate titles and captions which clearly describe the content without the need to refer to the text

Figure 1 – the inclusion of the figure is a helpful one but as CONSORT is designed to report RCTs and that we request reporting by STROBE guidelines for this observational study, suggest altering the title of the figure to “Participant inclusion, exclusion and matching” or something similar

Tables – we note the statistical reviewer’s comments (below) regarding the tables which we agree with.

Currently it is a journal requirement to present both 95% CIs and p-values when reporting main outcome measures and we therefore ask that you please report both rather than the use of bold and/or italic fonts to indicate significance 

Table 3 and 4 require revision for the purpose of improved accessibility – please apply the following to all tables presented in the main manuscript and as supporting information 

- Please revise to ensure that whole words are not split across lines – you may need to create the tables in landscape format to ensure that you have room, or the current tables can be split into more than 1

- Please ensure that confidence limits are not split across separate lines

- Please do not report p 0.000; report as p<0.001

- In an appropriate caption, please indicate which factors are adjusted for

- To help facilitate transparency of data reporting, please also provide the unadjusted analyses for comparison

DISCUSSION

Please remove the sub-heading “conclusions” form the end of the main manuscript such that the discussion reads as a single continuous piece of prose ending in a one paragraph conclusion

Please remove the funding statement, COI statement form the end of the main manuscript and include only in the manuscript submission form

REFERENCES

For in-text reference call-outs, citations should be placed within square brackets preceding punctuation as follows: “…reference [1,2].” Please note the absence of a space between citations. 

In the bibliography, for each reference listed, please list up to, but no more than, 6 author names followed by et al, where more than 6 authors contribute to the published article.

Journal name abbreviations should be those found in the National Center for Biotechnology Information (NCBI) databases. 

Please see our website for other reference guidelines: https://journals.plos.org/plosmedicine/s/submission-guidelines#loc-references

SUPPORTING INFORMATION

Please provide titles and legends for each individual table and figure in the Supporting Information.

eFigure 1 – please consider avoiding the use of green (or red) to improve accessibility to those with colour blindness

eTable 1 – as for the main manuscript tables, please ensure that words are not split across lines – see the column headers. Please also ensure that any and all abbreviations are defined in an appropriate caption (COPD, AMI, for example)

eTable 2 – we agree with the reviewer that modification of this table consistent with table 1 of the main manuscript

Comments from the reviewers:

Reviewer #1: Overall Impression:

This manuscript addresses an important gap in the scientific literature on the topic of long COVID, specifically among the elderly population. The authors accurately describe the challenges of diagnosing long COVID and identifying it in medical administrative data and highlight challenges among the elderly. The authors use methods similar to other studies of analyzing ICD-10 codes in medical claims data to identify possible cases of long COVID but apply them to a Medicare dataset and also include an influenza comparator group. The manuscript would advance our knowledge in ways to identify long COVID cases among this Medicare population and would be of interest to clinicians caring for this patient population. Overall, the conclusions are supported by the findings and in line with what has been previously reported in the general population. 

Major issues:

The use of the term "long flu" is a bit of a distraction. What the authors are trying to do with the influenza comparator group is valid and informative, but it seems less about trying to identify "long flu" (which also seems to have little scientific literature supporting it) and more about comparing the post-influenza time period with the post-COVID time period with respect to diagnoses/symptoms and healthcare utilization. I recommend the authors re-frame how they describe this aspect of the analysis and use a term like "influenza comparator group" or something similar. 

Additionally, I would like to see more in the discussion about how this analysis might change if done with more recent data after the introduction of the new Post-COVID condition ICD-10 code U09.9. While long COVID is still likely underdiagnosed, the introduction of this code will improve the coding of the diagnosis compared to when only B94.8 was available. 

Minor issues:

In the introduction, 3rd sentence, please include post-COVID condition or post-COVID conditions as these are the terms used by the U.S. Centers for Disease Control and Prevention and WHO. 

At the end of the first paragraph of the intro, suggest addition of the following reference: Point Prevalence Estimates of Activity-Limiting Long-term Symptoms Among United States Adults ≥1 Month After Reported Severe Acute Respiratory Syndrome Coronavirus 2 Infection, 1 November 2021 | The Journal of Infectious Diseases | Oxford Academic (oup.com)

In the results, there are several sentences throughout that deviate from a strict reporting of results and that some would consider more appropriate for the discussion. For example, in section 3.2 with the sentence "We speculate…" and in 3.4 "This is probably an anomaly….". 

For Table 2, suggest specifying "any of 11 symptoms" and "any of 3 main symptoms" so reader is not confused that you must have all 11 or all 3. This is explained in the methods, but readers might appreciate a reminder when reviewing the table. 

Reviewer #2: Thanks for the opportunity to review your manuscript. My role is as a statistical reviewer, so my review concentrates on the study design, data, and analysis that are presented. I have put general questions first, followed by queries relevant to a specific section of the manuscript (with a page/paragraph reference).

The rationale behind the work is interesting. People are (rightfully IMO) worried about long COVID symptoms and there has been research which has attempted to objectively look at what symptoms are more common following acute COVID infection. Longer-term effects from viral infection aren't unknown, but there hasn't (to my knowledge) been any attempts to understand which symptoms are also more common after influenza infection in the same rigorous way that COVID has been examined. Data from the USA Medicare program was taken from 2016-2021, limited to those >65 years, with enough data for a look-back period for pre-existing symptoms, not from a skilled nursing facility, and without a diagnosis for both COVID and influenza. Covid exposure was defined as unexposed, out-patient, and in-patient. Examination of long-covid was based on a specific ICD10 code, and the WHO symptom-based definition (based on newly reported symptoms that were unrelated to a recorded comorbidity). The symptom based definition was assessed by comparing patients exposed to COVID-19 to those without a record of exposure. The 'long COVID' symptom based definition was applied to patients with a diagnosis of influenza over the 2017/2018 flu seasons. Use of healthcare services was compared between patients with 'long COVID' and 'long Flu'. A sensitivity analysis using only the three most common long-COVID symptoms was also included. 

With the inclusion/exclusion criteria, and the temporal window of the data the results are specific to older people using Medicare, without records of both influenza and COVID. Some of the language in the conclusions is fairly general ('elderly COVID-10 outpatients and inpatients'), I wanted to check whether the exclusions necessary to create an appropriate cohort for this analysis might limit the generalisability of the results. 

P5, Paragraph. Given that PLOS Medicine has a global audience, an extra sentence or two describing the US Medicare system would be helpful, i.e. who is eligible to be enrolled, what does it cover?

Do all >65 year olds use Medicare and have their health services recorded in the database? Or do some in this age group pay for healthcare through other means (private insurance, out of pocket) that means part of the >65 year old population has no or incomplete records of health service use? 

P6, Paragraph 2. I wasn't sure how a SNFs differ from a hospital, or an aged care facility (nursing home/facility). Would this be a facility that provides care to less complex patients (i.e. a lower level hospital)? 

P8, Paragraph 5. Was testing in the US at this sufficiently complete, i.e. were all COVID cases likely to have the diagnosis recorded in the Medicare records? Or could the 'unexposed' include patients who had COVID but didn't receive a diagnosis?

P10, Paragraph 2. How were these covariates selected to include in the adjusted logistic regression model? 

P10, Paragraph 2. How was overdispersion in the Poisson model checked? Was the form of the residuals checked and ok? 

P11, Paragraph 2/Supp e Table 3. I wonder if standardised difference (often used with propensity score matching) would be an alternative to p-values here given the generous sample size you have. 

Also, to clarify does the unmatched rate refer to COVID patients unable to be matched?

P11, Paragraph 3. Just wondering if use of the B94.8 changed over time (as long COVID become more recognised)? No need to run additional analyses, just curious and would be interested to see if you have the output already. 

P12, Paragraph 2. The ED result is interesting, I would move the explanation to the discussion rather than having it in the results.

Table 3. Rather than using a significance level (bolding) it would be fine to just have the p-values and ORs displayed here (and in other tables). I would also modify the p-values listed as "0.000" to be "<0.001". With the sample size, reporting the percent with the symptom to two decimal places is probably more than needed as well, reporting to just percent (or one decimal place where the % is <1) would be fine here. 

eFigure 1. I would update the Y-axis to remove the extra 00s in percent, i.e. 25% instead of 25.00% The dots also overlap and it is difficult to see the individual series in the neuropsychiatric graph, I would suggest expanding the Y scale and maybe also using lines instead of dots so there is less overlap. 

Etable 2. I would modify the table so it's clear which categories belong to each variable, e.g. it wasn't clear that 'other' and 'northeast' belonged to separate variables initially. The way it is laid out in Table 1 (main manuscript) is good. 

Reviewer #3: Publications on COVID-19 sequelae in the elderly population is largely lacking, and the authors of this manuscript are commended for focusing on this age group. The authors utilize Medicare data, and compare post-covid and post-flu registered diagnoses and symptoms captured in medical records during the period April 2020 to June 2021. Diagnosis-based approach yielded low estimated incidence, while the symptom approach gave higher estimates. This concurs with previous reports showing that specific diagnosis-based estimates likely underestimate true prevalence to a large extent. While the same is likely true also for specific symptom reporting, this paper reports as high as 17% post infection symptoms both after COVID and influenza in outpatient settings and 25-29% in hospitalized settings.

Some questions remain:

1. Comorbidities are known risk factors for long COVID. Still, the authors excluded patients with encounters during the past year. What encounters are considered here? If it means any contact with the health services, this seems too strict and will exclude many individuals with risk of prolonged symptoms. Did you look at data without this exclusion criteria? 

2. In table 3 individual symptoms are compared. Dyspnoea and fatigue are as expected high post COVID, while memory or cognitive symptoms are at a lower frequency that have previously been reported in prospective cohorts of younger age. At the same time increased incidence of dementia has been reported in this age group, as well as MRI changes in COVID patients. Could the threshold for patients, as well as doctors, reporting of this symptom be higher, for various reasons, and therefore the approach in this study for cognitive symptoms be less sensitive? 

3. The data on flu are an important addition to the present literature on flu complications.

4. In the discussion it is stated that long COVID symptoms in 16% of outpatients and 29% of hospitalised patients is similar to published data. Even though there is a large literature here, and perhaps hard to keep an overview, the references given (refs 8 through 19), should have been penetrated more in order to support the claim. The authors have not attempted to discriminate between hospitalised and non-hospitalised cohorts, and even included population-based health record data, based on similar approaches as the diagnosis-based which the authors show underestimate prevalence in this manuscript. 40% to 70% post COVID symptoms would be more appropriate when scrutinizing existing literature, particularly after hospitalisations. This means that even the approach in this manuscript under-estimates the incidence of long COVID symptoms. This needs to be acknowledged in the discussion. It doesn't make the data less interesting, since this manuscript likely shows an approach which is better suited than previous diagnosis-based approaches, and can be seen as an improved long COVID estimate based on large populations.

[LINK]

---

## [Decision Letter · Decision Letter 2]

3 Mar 2023

Dear Dr. Fung,

Thank you very much for re-submitting your manuscript "Long COVID in Elderly Patients: An Epidemiologic Exploration Using a Medicare Cohort" (PMEDICINE-D-22-03023R2) for review by PLOS Medicine.

I have discussed the paper with my colleagues and the academic editor and it was also seen again by 2 reviewers. I am pleased to say that provided the remaining editorial and production issues are dealt with we are planning to accept the paper for publication in the journal.

[LINK]

We look forward to receiving the revised manuscript by Mar 24 2023 11:59PM.   

Sincerely,

Philippa Dodd, MBBS MRCP PhD

PLOS Medicine

plosmedicine.org

Requests from Editors:

GENERAL

Thank you for considered and detailed responses to editor and reviewer comments.

Please see below for further minor points that we request you respond to in full.

DATA AVAILABILITY STATEMENT

Thank you for your detailed statement. Please include a URL or contact email address for the CMS Virtual Research Data Center.

TITLE

We wonder if your title could better reflect your study design and size. Suggest revising in line with our guidance. Your title must be nondeclarative and not a question. It should begin with main concept if possible. "Effect of" should be used only if causality can be inferred, i.e., for an RCT. Please place the study design ("A randomized controlled trial," "A retrospective study," "A modelling study," etc.) in the subtitle (ie, after a colon).

INTRODUCTION

Thank you for modifying your introduction to define long-flu more clearly.

Line 118 suggest, “This means that some patients would satisfy the diagnostic criteria for long COVID after an episode of influenza had the primary infection been COVID-19 instead. For lack of a better name, we shall call this condition “long Flu”.

ABSTRACT

Throughout you report long flu in different ways “long Flu, long flu and long-flu”

Similarly “long-COVID Vs long COVID” is also written please amend for consistency, including throughout the rest of the manuscript.

AUTHOR SUMMARY

This reads very nicely but is a little too long. Ideally each sub-heading should contain 2-3 single sentence, concise bullet points. Please revise including the most salient points from your study.

TABLES

Table 1: We previously requested that the presentation of p values in the tables were revised but in my version they appear unchanged. Please report p as p<0.001 or where higher as p=0.002, for example (not p<.0…)

Table 2: as above, please revise the presentation of p values. Please also report Please ensure that numerical values contained in brackets are clearly defined for the reader to ensure accessibility – this is not evident in some cases i.e., where (presumably) p values are presented). 

Table 3: Thank you for indicating that your analyses are adjusted, we previously asked you to include unadjusted analyses for comparison but could not find these. For the purpose of transparent data reporting please include the unadjusted analyses. If not please clearly state the reasons why not.

Table 4: Thank you for indicating that your analyses are adjusted, we previously asked you to include unadjusted analyses for comparison but could not find these. For the purpose of transparent data reporting please include the unadjusted analyses. If not please clearly state the reasons why not.

Row header: “Comparative risk of long COVID patients relative to long Flu patients” – we wonder whether a different header could better describe these data? Suggest “Comparative risk in long COVID compared to long flu” or similar?

DISCUSSION

Please ensure that you present and organize the Discussion as follows: a short, clear summary of the article's findings; what the study adds to existing research and where and why the results may differ from previous research; strengths and limitations of the study; implications and next steps for research, clinical practice, and/or public policy; one-paragraph conclusion. 

REFERENCES

For in-text reference callouts, please remove spaces from between citations. For example line 93 should read “[22,23].”

SOCIAL MEDIA

To help us extend the reach of your research, please provide any Twitter handle(s) that would be appropriate to tag, including your own, your co-authors’, your institution, funder, or lab. Please detail any handles you wish to be included when we tweet this paper, in the manuscript submission form when you re-submit the manuscript.

Comments from Reviewers:

Reviewer #2: Thanks for the revised manuscript and responses to my review. The revision resolves the queries from my first review. The Medicare population is clearly explained, and I think the limitations of using this data are articulated in the discussion. 

Reviewer #3: The authors have responded well to all my queries, and I think the manuscript can be published.

[LINK]

---

## [Editor Report · Decision Letter 3]

14 Mar 2023

Dear Dr Fung, 

On behalf of my colleagues and the Academic Editor, Professor Mirjam Kretzschmar, I am pleased to inform you that we have agreed to publish your manuscript "Prevalence and Characteristics of Long COVID in Elderly Patients: An Epidemiologic Exploration Using a Retrospective Medicare Cohort" (PMEDICINE-D-22-03023R3) in PLOS Medicine.

Before we can publish your manuscript we request that you make the following changes:

1) TITLE - suggest the following, or similar:

Prevalence and characteristics of long COVID in elderly patients: An observational cohort study of over 2 million adults in the US

2) AUTHOR SUMMARY

Suggest combining points on lines 60 and 63 into one to read as follows:

“Despite the similar overall incidence rates, long COVID patients suffered with notably different symptoms compared to long Flu patients and were also more likely to access inpatient and outpatient healthcare services.” 

Line 66: suggest “The use of designated long COVID diagnostic codes alone is likely to result in gross under reporting of long COVID in this population."

Line 68: please remove this statement as it repeats the findings listed above

Line 71: suggest “Long COVID is associated with greater healthcare utilization than long Flu, suggesting a bigger impact on individual health and well-being, as well as on healthcare expenditure.

3) DISCUSSION

Line 422: “However, we can venture…” suggest making this a separate paragraph

PRESS

Best wishes,

Pippa 

Philippa Dodd, MBBS MRCP PhD 

PLOS Medicine